# Iron Ions Increase the Thermal Stability In Vitro and Activity In Vivo of the 447R Mutant Form of Mouse Tryptophan Hydroxylase 2

**DOI:** 10.3390/ijms26178188

**Published:** 2025-08-23

**Authors:** Polina D. Komleva, Ekatherina I. Terentieva, Arseniy E. Izyurov, Alexander V. Kulikov

**Affiliations:** The Federal Research Center Institute Cytology and Genetics, Russian Academy of Sciences, Novosibirsk 630090, Russia; komleva@bionet.nsc.ru (P.D.K.); terentevaei@bionet.nsc.ru (E.I.T.); izyurovae@bionet.nsc.ru (A.E.I.)

**Keywords:** pharmacological chaperone, iron ions, Fe(III) hydroxide dextran complex, tryptophan hydroxylase 2, P447R mutation, thermal stability, brain, mice

## Abstract

Tryptophan hydroxylase 2 (TPH2) hydroxylates L-tryptophan to L-5-hydroxytryptophan (5-HTP) the first and rate-limiting step of serotonin (5-HT) synthesis in the mammalian brain. Some mutations in the *Tph2* gene reducing TPH2 activity are associated with hereditary depressive disorders. The P447R substitution in the mouse TPH2 molecule reduces its thermal stability in vitro and its activity in the brain. The effects of iron ions on thermal stability in vitro and the activity in the brain of the mutant TPH2 were investigated. In the in vitro experiment effects of 0.01, 0.05, and 0.2 mM of FeSO_4_ and FeCl_3_ on the enthalpy (ΔH) and Gibbs free energy (ΔG) of thermal denaturation of the mutant TPH2 extracted from the midbrain of Balb/c mice were assayed. All FeSO_4_ concentrations and 0.05 and 0.2 mM concentrations of FeCl_3_ increased these thermodynamic characteristics of the mutant TPH2. Repeated (for 7 days) intramuscular administration of Fe(III) hydroxide dextran complex (15 and 30 mg/kg/day) increased TPH2 activity in the hippocampus, but not in the midbrain in Balb/c mice. Repeated (for 7 days) intramuscular administration of Fe(III) hydroxide dextran complex (15 and 30 mg/kg/day) together with thiamine (8 mg/kg/day) and cyanocobalamin (0.8 mg/kg/day) increased TPH2 activity in the hippocampus, while 30 mg/kg of Fe(III) hydroxide dextran also increased the enzyme activity in the midbrain in Balb/c mice. These results are the first evidence for chaperone-like effects of iron ions on thermal stability in vitro and activity in the brain of the mutant TPH2.

## 1. Introduction

Tryptophan hydroxylase 2 (TPH2) hydroxylates L-tryptophan to L-5-hydroxytryptophan (5-HTP) and is the first and rate-limiting step of serotonin (5-HT) synthesis in the mammalian brain [1,2,3,4,5]. Some mutations in the *Tph2* gene reducing TPH2 activity are associated with hereditary depressive disorders [6,7,8]. The P447R substitution in the mouse TPH2 molecule reduces its activity in the mouse brain [9,10] and its thermal stability in vitro [11,12]. Although there are several TPH2 irreversible and concurrent TPH2 inhibitors, drugs that normalize the mutant enzyme activity are still an unresolved fundamental and medical problem of modern psychopharmacology [13].

Pharmacological chaperones are chemicals that restore the thermal stability in vitro and the functional activity in vivo of mutant molecules. They are considered as promising drugs for pharmacological treatment of some grave hereditary diseases [14,15,16,17,18]. TPH2 is a member of the family of aromatic amino acid hydroxylases including tryptophan hydroxylase 1, tyrosine, and phenylalanine hydroxylases [19]. The temperature at which half of the studied protein is denatured (T_50_) is a widely used index of its thermal stability in vitro. The higher T_50_, the higher the protein thermal stability [20]. The natural cofactor for all the aromatic amino acid hydroxylases, thetrahydrobiopterine (BH_4_), was shown to increase the T_50_ of the thermal denaturation of tryptophan, tyrosine and phenylalanine hydroxylases in vitro [11,20,21,22,23,24,25,26]. At the same time, exogenous BH_4_ fails to increase TPH2 activity in vivo [11].

Recently, we found that Fe(II) and Fe(III) ions increased T_50_ of TPH2 thermal denaturation in vitro [27]. However, only the ligand-induced increase in the energy of thermal denaturation is the strong evidence for its chaperone-like function. Moreover, taking into consideration the chaperone-like activity of iron ions in vitro, we hypothesized that these ions can stabilize mutant TPH2 and increase its total activity in the mouse brain (in vivo).

In this study, this idea was further developed and the effects of (1) Fe(II) and Fe(III) ions on the thermodynamics of thermal denaturation in vitro and (2) repeated intramuscular administration of Fe(III) hydroxide dextran complex on the mutant TPH2 activity in mouse midbrain (where the bodies of 5-HT neurons are located and TPH2 is synthesized) and hippocampus (where the endings of 5-HT neurons are located) in vivo were investigated. Here, we intended to test whether Fe(II) and Fe(III) ions increased the energy (enthalpy ΔH and Gibbs free energy ΔG) of denaturation in vitro, and whether a repeated administration of exogenous iron complex increased the activity of mutant TPH2 in the midbrain and hippocampus (in vivo).

## 2. Results

### 2.1. Effects of FeSO_4_ and FeCl_3_ on ΔH, ΔS, and ΔG of the Thermal Denaturation of the Mutant TPH2 In Vitro

All studied concentrations of FeSO_4_ (Fe(II)) increased in vitro ΔH (F(3,21) = 79.01, *p* < 0.001), ΔS (F(3,21) = 80.0, *p* < 0.001), and ΔG (F(3,21) = 64.9, *p* < 0.001) the thermal denaturation of the mutant TPH2 extracted from the Balb/c midbrain (Figure 1).

The FeCl_3_ (Fe(III)) dose-dependently increased in vitro ΔH (F(3,24) = 17.95, *p* < 0.001), ΔS (F(3,24) = 17.69, *p* < 0.001), and ΔG (F(3,24) = 19.67, *p* < 0.001) of the thermal denaturation of the mutant TPH2 extracted from the Balb/c midbrain. However, it increased these characteristics only at concentrations 0.05 and 0.2 mM, but not 0.01 mM (Figure 2).

Comparison of the increase in ΔH, ΔS, and ΔG (ΔΔH, ΔΔS, ΔΔG) caused by FeSO_4_ and FeCl_3_ over their basal levels using a two-way ANOVA reveals the effects of iron ion type and concentration factors, but not these factors interaction (Table 1).

### 2.2. Effect of Repeated Intramuscular Administration of Fe(III) Hydroxide Dextran Complex on the TPH2 Activity in the Midbrain and Hippocampus and the Tph2 Gene Expression in the Midbrain in Balb/c Mice

The Fe(III) hydroxide dextran complex at the concentration of 0.1 mM did not affect T_50_ (F(1,14) < 1), ΔH (F(1,14) < 1), ΔS (F(1,14) < 1), and ΔG (F(1,14) < 1) in the thermal denaturation of the mutant TPH2 extracted from the midbrain of Balb/c mice in vitro (Figure 3).

Intramuscular administration of 15 and 30 mg/kg of Fe(III) hydroxide dextran complex for 7 successive days increased TPH2 activity in the hippocampus (F(2,26) = 5.69, *p* = 0.009, Figure 4), but not in the midbrain (F(2,26) < 1, Figure 4) in Balb/c mice. None of these doses affected the level of *Tph2* gene mRNA in the midbrain (F(2,25) = 1.66, *p* = 0.21, Figure 4) in Balb/c mice.

Intramuscular administration of 15 and 30 mg/kg of Fe(III) hydroxide dextran complex together with 8 mg/kg of thiamine and 0.08 mg/kg of cyanocobalamin for 7 successive days also increased TPH2 activity in the hippocampus (F(3,28) = 7.69, *p* < 0.001, Figure 5). Moreover, 30 mg/kg of Fe(III) hydroxide dextran complex together with the vitamin mix increased the activity of TPH2 (F(3,28) = 3.18, *p* < 0.039, Figure 5) and the level of *Tph2* gene mRNA in the midbrain (F(3,27) = 4.06, *p* < 0.017, Figure 5). At the same time, the vitamin mix itself did not affect the activity of TPH2 in both brain structures and the level of *Tph2* gene mRNA in the midbrain (Figure 5).

## 3. Discussion

The main aims of this study were to find out (1) whether Fe(II) and Fe(III) ions increased the energy (ΔH, ΔG) of thermal denaturation in vitro, and (2) whether a repeated administration of Fe(III) hydroxide dextran complex increased the activity of mutant TPH2 in mouse brain (in vivo).

The first aim of this study was to find strong evidence that Fe(II) and Fe(III) ions increased the energy (ΔH, ΔG) of thermal denaturation in vitro. The P447R substitution in mouse TPH2 molecules decreases the Gibbs free energy (ΔG) of its thermal denaturation [12]. Recently, we showed that both Fe(II) and Fe(III) ions increased the T_50_ of thermal denaturation the mutant mouse TPH2 in vitro [27]. Many authors consider the ligand-induced increase in T_50_ as sufficient evidence of its chaperone-like function [20,21,22,23,24,25]. However, only the ligand-induced increase in the energy of thermal denaturation is strong evidence for a ligand chaperone-like function. In this study, we first demonstrated that Fe(II) and Fe(III) ions increased the ΔH and ΔG of the mutant TPH2 thermal denaturation. This result provides strong direct evidence of a chaperone-like function of iron ions: they interact with the mutant TPH2 molecule and increase the energy of its thermal denaturation.

The Fe(II) ion links to the catalytic center of TPH2 and it is mandatory for the hydroxylation of L-tryptophan [19,28]. This well-known catalytic function of iron ions should be distinguished from its chaperone function discovered in this study: these two functions of iron ions are evaluated by the effects of Fe(II) on the activity and the thermal denaturation of the TPH2 molecule, correspondingly. The absence of statistical effect of the interaction of iron ion type x concentration on the thermodynamic characteristics of TPH2 thermal denaturation indicates that both Fe(II) and Fe(III) similarly stabilized the mutant TPH2 molecule. However, it cannot be ruled out that in the incubation environment Fe(II) can be oxidized to Fe(III) and vice versa and there are a balance between them. Further studies on recombinant proteins are necessary to resolve the problem of the ratio of chaperone activities of Fe(II) and Fe(III) ions.

A question arises: what is the hypothetical molecular mechanism of the observed chaperone-like effect of iron ions? Iron ions can coordinate with histidine, glutamate, and aspartate residues of TPH2 molecule as the ligands. For example, in the catalytic center of human TPH2, Fe(II) forms dative bonds with H318, H323, and G363 residues. The total energy of these bonds is high enough to stabilize the catalytic center. It cannot be also ruled out that Fe(II) and Fe(III) could stabilize the TPH2 molecule via bonds with other sets of histidine, glutamate, and aspartate residues. 

Recently it was shown that Mn(II) ion also increased in vitro the T_50_ of TPH extracted from zebrafish brain [29]. It is well known that Mn(II), Fe(II) as well as ions of some transition metals can coordinate histidine, glutamate and aspartate residues [30]. Maybe ions of some transition metals also can increase of the thermal stability of TPH2 molecule. In the present study we only put the problem and many efforts need to solve it.

Taking into consideration the chaperone-like activity of iron ions in vitro, we hypothesized that their increased concentration in the cytoplasm of 5-HT neurons can stabilize mutant TPH2, increase its life span and its total activity in the mouse brain. The second aim of this study was to test this hypothesis and answer the following question: whether a repeated administration of exogenous iron ions increase the activity of mutant TPH2 in the mouse brain or not. For this purpose, we used Fe(III) hydroxide dextran complex in its pharmacological form for intramuscular administration, since it is the most effective pharmacological means for anemia treatment and it does not include any additional chemicals. This drug does not penetrate into the brain and does not affect the thermal denaturation of the mutant TPH2 in vitro. At the same time, the organism has some mechanisms that digest dextran and release Fe(III) which then links with transferrin, forming the transferrin-2Fe(III) complex.

A mechanism for the penetration of Fe(III) through the blood–brain barrier includes (1) the binding of the transferrin-2Fe(III) complex to the transferrin-binding receptor 1 at the luminal pole of endothelial cells of blood vessels, (2) the internalization of the transferrin-2Fe(III)–transferrin-binding receptor 1 complex into the endosomes, (3) the fusion of the endosomes with the abluminal membrane of the endothelial cell, and (4) the liberation of the transferrin-2Fe(III) complex into the interstitial space [31]. The transferrin-2Fe(III) complex from the interstitial space enters the neuron by fusion with the transferrin-binding receptor 1. In the neuron, Fe(III) is reduced to Fe(II) [32].

The treatment duration (7 days) was chosen as a compromise for two reasons. First, some time is required to accumulate enough iron ions in the cytoplasm of 5-HT neurons and to replace less stable TPH2 with the more stable TPH2-iron complex. The half-life for TPH2 in the brain is about 48 h [33,34]. Second, we reduced the number of intramuscular injections in order to minimize discomfort for an animal.

We found that 7 days of intramuscular administration of 15 and 30 mg of Fe(III) hydroxide dextran complex was sufficient to increase the mutant TPH2 activity in the hippocampus. Since the drug administration does not increase of the *Tph2* gene expression (in the midbrain), the observed drug-induced increase in the TPH2 activity in the hippocampus seems to result from an increase of the enzyme stability rather than its synthesis de novo. At the same time, both doses of Fe(III) hydroxide dextran failed to alter the TPH2 activity in the midbrain. One possible explanation why Fe(III) hydroxide dextran increases TPH2 activity in the hippocampus, but not in the midbrain is that more exogenous Fe(III) penetrates in the hippocampus than in the midbrain, since the total surface of the 5-HT ending in the hippocampus is bigger than that of the 5-HT neuron bodies in the midbrain raphe nuclei. We could not test this hypothesis due to lack of a sensitive and reliable technique for quantitative assay of iron ion content in the neuron’s cytoplasm.

Since thiamine and cyanocobalamin facilitate penetration and accumulation of iron ions in the organism, and they are frequently used as adjuvants to iron complexes for anemia treatment, it was hypothesized that these vitamins could increase the effect of Fe(III) hydroxide dextran on the TPH2 activity. In order to test this hypothesis, mice were intramuscularly treated with Fe(III) hydroxide dextran together with these vitamins. As in the antecedent in vivo experiment, a marked increase in the TPH2 activity in the hippocampus of the animals treated for 7 days with both doses of Fe(III) hydroxide dextran together with these vitamins was revealed. Moreover, unlike the antecedent in vivo experiment, in this experiment, a marked increase in TPH2 activity in the midbrain in mice treated with high-dose of Fe(III) hydroxide dextran together with the vitamins was observed. This increase in the TPH2 activity in the midbrain was accompanied with a marked increase in the *Tph2* gene expression in this structure. Therefore, the observed increase in the TPH2 activity in the midbrain after repeated administration of Fe(III) hydroxide dextran together with the vitamins could result from not only an increase in enzyme stability, but also from TPH2 synthesis de novo. This increase in the *Tph2* gene expression in the midbrain after treatment with a high dose of Fe(III) hydroxide dextran together with vitamins has been observed in the first time and its molecular mechanism is still unknown.

## 4. Materials and Methods

### 4.1. Animals

All experiments were carried out on 12 month-old males of Balb/c mice (*n* = 82). During the experiments, the animals had specific pathogens free state, were kept separately in cages (Optimice, Animal Care Systems, Centennial, CO, USA) at the temperature 24 ± 2 °C, humidity 45–50%, and artificial 14:10 (light: dark) photoperiod with daybreak and sunset at 01:00 and 15:00, respectively, fed with sterile food and water ad libitum.

### 4.2. Chemicals

Stock solutions of FeSO_4_ 7H_2_O (25 mM) (Sigma-Aldrich, Darmstadt, Germany) and FeCl_3_ 6H_2_O (Sigma-Aldrich, Darmstadt, Germany) (25 mM) in ultrapure water were used as sources of Fe(II) and Fe(III) ions in the in vitro experiment (experiment 1).

The Fe(III) hydroxide dextran solution (50 mg Fe(III)/mL) (Sandoz, Ljubljana, Slovenia) was used in all in vivo experiments (experiments 2–3). Thiamine hydrochloride (50 mg/mL) (Borimed, Borisov, Belarus) and cyanocobalamin (0.5 mg/mL) (Mosagrogene, Moscow, Russia) were used in the experiment 3.

### 4.3. Experiments

Experiment 1. The effect of Fe(II) and Fe(III) ions as well as Fe(III) hydroxide dextran on the thermodynamic characters of thermal denaturation of the mutant TPH2 in vitro was examined. Twenty Balb/c males were euthanized with CO2, decapitated, and each of their 20 midbrains were homogenized in 500 µL of cold 50 mM Tris HCl buffer, pH 7.6, containing 1 mM of dithiothreitol (Sigma-Aldrich, Darmstadt, Germany), spun for 15 min at 24,700 rpm (+4 °C). The clear supernatants were pulled, aliquoted by 300 µL and stored at −80 °C as source of mutant TPH2 for studying the effects of various concentrations of FeSO_4_, FeCl_3_, and Fe(III) hydroxide dextran on the thermodynamic characters of the thermal denaturation in vitro.

In this study, thermal denaturation curves were constructed for the mutant form of TPH2 in the presence of 0.05, 0.1, and 0.2 mM FeSO_4_ and FeCl_3_ or 0.1 mM Fe(III) hydroxide dextran. Thermal denaturation curves without the addition of iron ions served as controls. The T_50_, ΔH, ΔS, and ΔG were then calculated from the denaturation curves. Four or five thermal denaturation curves were constructed for each iron ion concentration, Fe(III) hydroxide dextran, and the control.

Experiment 2. The effect of repeated intramuscular administration of Fe(III) hydroxide dextran on the TPH2 activity and the *Tph2* gene expression was analysed. Thirty adult Balb/c males were divided into three groups: control (*n* = 10), 15 mg/kg (*n* = 10), and 30 mg/kg (*n* = 10) of Fe(III) hydroxide dextran. Aliquots of 300 (15 mg/kg) and 600 (30 mg/kg) µL Fe(III) hydroxide dextran were transferred to clean Eppendorf tubes (2 mL) and made up to 2 mL with sterile saline. The solutions were administered intramuscularly in a hind leg once a day at a volume of 2 µL/g body mass for 7 successive days. The control animals were treated intramuscularly for 7 days with saline. On the 8th day, the animals were euthanized with CO_2_, decapitated, and the midbrain and hippocampus were isolated, frozen with liquid nitrogen, and stored at −80 °C.

Experiment 3. The effect of repeated intramuscular administration of Fe(III) hydroxide dextran together with thiamine hydrochloride and cyanocobalamin on the TPH2 activity and the *Tph2* gene expression was examined. Thirty-two adult Balb/c males were divided into four groups: control (*n* = 8), vitamins (*n* = 8), vitamins with 15 mg/kg (*n* = 8), and vitamins with 30 mg/kg (*n* = 8) of Fe(III) hydroxide dextran. Aliquots of 150 µL thiamine hydrochloride (8 mg/kg) and 150 µL cyanocobalamin (0.08 mg/kg) were transferred into three Eppendorf tubes. In the second and the third tubes, 300 µL (15 mg/kg) and 600 µL (30 mg/kg) Fe(III) hydroxide dextran were also transferred and all tubes were made up to 2 mL with sterile saline. The solutions were administered intramuscularly in a hind leg once a day at a volume of 2 µL/g body mass for 7 successive days. The control animals were treated intramuscularly for 7 days with saline. On the 8th day, the animals were euthanized with CO_2_, decapitated, and the midbrain and hippocampus were isolated, frozen with liquid nitrogen, and stored at −80 °C. These two structures were chosen as the places of the bodies and endings of 5-HT neurons, respectively.

The doses of 15 and 30 mg/kg of Fe(III) hydroxide dextran in the 2nd and 3rd experiments are in the range of doses (15–75 mg/kg) of this drug used for administration to mice [35]. The doses of thiamine hydrochloride (8 mg/kg) and cyanocobalamin (0.08 mg/kg) in experiment 3 were chosen in accordance with veterinarian instructions for treatment of cats.

In experiments 2 and 3, the midbrain and hippocampus were homogenized in 400 and 300 µL, correspondingly, of cold 50 mM Tris HCl buffer, pH 7.6, containing 1 mM of dithiothreitol. Aliquots of 100 µL of homogenates from the midbrain was mixed with 1 mL of ExtractRNA reagent (Eurogene, Moscow, Russia) for total RNA extraction (see 4.6). Then, the homogenates from the midbrain and hippocampus were spun for 15 min at 24,700 rpm (+4 °C). The clear supernatant was collected in clean tubes and stored at −80 °C until the TPH2 activity assay.

### 4.4. TPH2 Activity Assay

Aliquots of 15 µL of clear supernatant was incubated for 15 min at 37 °C with 0.3 mM of L-tryptophan (Sigma-Aldrich, Darmstadt, Germany), 0.3 mM of BH_4_ (Sigma-Aldrich, Darmstadt, Germany), 0.3 mM of m-hydroxybenzylhydrazine (Sigma-Aldrich, Darmstadt, Germany), and 5 U of catalase (Sigma-Aldrich, Darmstadt, Germany) in the final volume of 25 µL. The reaction was terminated by 75 µL of 0.6 M HClO_4_ and spun for 15 min at 24,700 rpm (+4 °C). Aliquots of 80 µL of the supernatant were mixed with 80 µL of ultrapure water and the synthesized 5-HTP concentration was assayed using HPLC on a Luna C18(2) column (5 μm particle size, L × I.D. 100 × 4.6 mm, Phenomenex, Torrance, CA, USA) with electrochemical detection (750 mV, DECADE II™ Electrochemical Detector; Antec, Alphen aan den Rijn, The Netherlands), a glassy carbon flow cell (VT-03 cell 3 mm GC sb; Antec, The Netherlands), CBM-20 A system controller, LC-20AD solvent delivery unit, SIL-20 A autosampler, and DGU-20A5R degasser (Shimadzu Corporation, Kyoto, Japan). The mobile phase (pH = 3.2) contained 13.06 g of KH_2_PO_4_, 200 μL 0.5 M Na_2_EDTA, 300 mg 1-octanesulfonic acid sodium salt (Sigma-Aldrich, Darmstadt, Germany), 940 μL concentrated H_3_PO_4_, and 130 mL methanol (13% volume; Vektor Ltd., Russia) in 1 L. The 5-HTP concentration was calibrated against the calibrated curves for corresponding standards of 25, 50, and 100 pmoles of 5-HTP (Sigma-Aldrich, Darmstadt, Germany). The TPH activity was measured as pmoles of 5-HTP formed per minute per mg of protein [12].

### 4.5. Assay of Fe(II), Fe(III) Ions and Fe(III) Hydroxide Dextran on TPH2 Thermal Stability In Vitro

Aliquots of 10 μL pure supernatant (see Section 4.3., experiment 1) were mixed with 5 μL 50 mM Tris HCl buffer (pH 7.6) containing 1 mM dithiothreitol or with 0.03, 0.15, 0.6 mM of FeSO_4_, FeCl_3_ (the final concentrations were 0.01, 0.05, and 0.2 mM) and 0.3 mM of Fe(III) hydroxide dextran (the final concentration was 0.1 mM) solutions in this buffer and heated for 2 min at 48, 50, 52, 54, 56, 58, 60, 62, and 64 °C and then cooled down in ice. The control tubes were not heated. Then, 10 μL of the mix of L-tryptophan (0.75 mM), 6-methyl-5,6,7,8-tetrahydropteridine (0.75 mM), m-hydroxybenzylhydrazine (0.75 mM) and catalase (5 U) in 50 mM Tris HCl buffer, pH 7.6, with 1 mM of dithiothreitol (the final concentrations of these chemicals were 0.3, 0.3, and 0.3 mM, respectively) was added, and the amount of synthesized 5-HTP was assayed using HPLC after a 15 min incubation at 37 °C (see Section 4.4). Eight groups of 4–5 thermal curves each were formed: (1) without iron ions, (2) with 0.01 mM of FeSO_4_, (3) with 0.05 mM of FeSO_4_, (4) with 0.2 mM of FeSO_4_, (5) with 0.01 mM of FeCl_3_, (6) with 0.05 mM of FeCl_3_, (7) with 0.2 mM of FeCl_3_, and (8) 0.1 mM of Fe(III) hydroxide dextran. The linear parts of these curves were used to calculate T_50_, ΔH, ΔS, and ΔG values using a linear regression method [11,12,27].

### 4.6. Assay of Tph2 Gene mRNA Level

Total mRNA was extracted from the homogenate with ExtractRNA reagent (see Section 4.3) according to the manufacturer’s protocol, treated with RNAase free DNAase (Promega, Medisson, WI, USA) according to the manufacturer’s protocol, and its concentration was diluted to the final concentration of 125 ng/μL. The cDNA was synthesized using a set of random hexanucleotide primers and R01 Kit according to the manufacturer’s protocol (Biolabmix, Novosibirsk, Russia). The mRNA level of target genes was assayed by qPCR using the set of selective primers (Table 2) and R401 Kit (Sintol, Moscow, Russia) according to the manufacturer’s protocol (95 °C 5 min; (95 ◦C, 15 s; annealing temperature, 60 s; 82 °C, 2 s; fluorescence registration) × 40 cycles). The threshold cycles were calibrated with a set of external standards containing 25, 50, 100, 200, 400, 800, 1600, 3200, and 6400 copies of genomic DNA extracted from C57BL/6 mouse liver. The gene expression was presented as a relative number of cDNA copies calculated on 100 copies of *Polr2a* cDNA as an internal standard [12].

### 4.7. Statistics

#### 4.7.1. Analysis of the Thermal Denaturation Curves [12]

For T_50_ calculation, the thermal denaturation curves in the coordinates T and (Vc − Vt)/Vc were used (T—Kelvin temperature °K of heating), Vt—the TPH2 activity after heating at temperature t, Vc—the TPH2 activity in the control sample). The linear part of each curve was approximated by equation T = b × (Vc − Vt)/Vc + a, and the b and a coefficients were calculated. Then, using these coefficients, the T_50_ value was calculated as T_50_ = b × 0.5 + a.

For calculation of the ΔH, ΔS, and ΔG, the equation ΔG = ΔH − ΔS × T was used. Where T = T_50_, ΔG = 0, and ΔH = ΔS/T_50_. The thermal denaturation curves in the coordinates (Vc − Vt)/Vc and T were plotted. The linear part of each curve was approximated by equation (Vc − Vt)/Vc = d × T + c and the d slope coefficient was calculated. Using the d and T_50_ values, ΔH = d × R × (T_50_)^2^ (R—gas constant, 1.987 cal/grad/mole) and ΔS = ΔH/T_50_ were calculated. Using these ΔH and ΔS values, the standard ΔG value (at 25 °C) was calculated according to the equation ΔG = ΔH − ΔS × 298.15.

#### 4.7.2. Statistical Tests

All data were presented as the mean ± SEM and analyzed using one-way ANOVA. Post hoc analyses were carried out using Fisher’s LSD multiple comparison test when appropriate. Statistical significance was set at *p* < 0.05.

## 5. Conclusions

Pharmacological chaperones are promising but still obscure pharmacological means for the treatment of inherited diseases caused by mutation misfolding of the target proteins [18]. Most authors search for potential pharmacological chaperones among complex organic chemicals. In this study using the mutant 447R form of mouse TPH2 as a model, as well as in vitro biochemical and in vivo pharmacological techniques, a chaperone-like activity of Fe(II) and Fe(III) ions was first investigated. Fe(II) and Fe(III) ions in concentration 10–200 μM increased the thermodynamic characteristics and ΔH and ΔG of thermal denaturation of the mutant form of TPH2 in vitro. Moreover, repeated intramuscular treatment with Fe(III) hydroxide dextran increased the activity of mutant TPH2 in the hippocampus, but not in the midbrain in Balb/c mice. Thiamine and cyanocobalamin seems to enhance the positive effect of intramuscular administration of Fe(III) hydroxide dextran on the activity of mutant TPH2 and an increase in the enzyme activity is observed also in the midbrain, not only in the hippocampus.

Two practically important consequences follow from the obtained results. First, the interaction of a protein molecule with iron ions is simpler than with any complex organic chemical and can be precisely calculated with some computer chemistry software (for example Gaussian). Therefore, the developed model opens up broad possibilities to understand the molecular mechanisms with the help of which pharmacological chaperones correct target protein molecule misfolding. Second, iron complexes can be effective for correcting some depressive disorders caused by mutations in the *Tph2* gene. However, the last conclusion needs further experimental and clinical study.

## Figures and Tables

**Figure 1 ijms-26-08188-f001:**
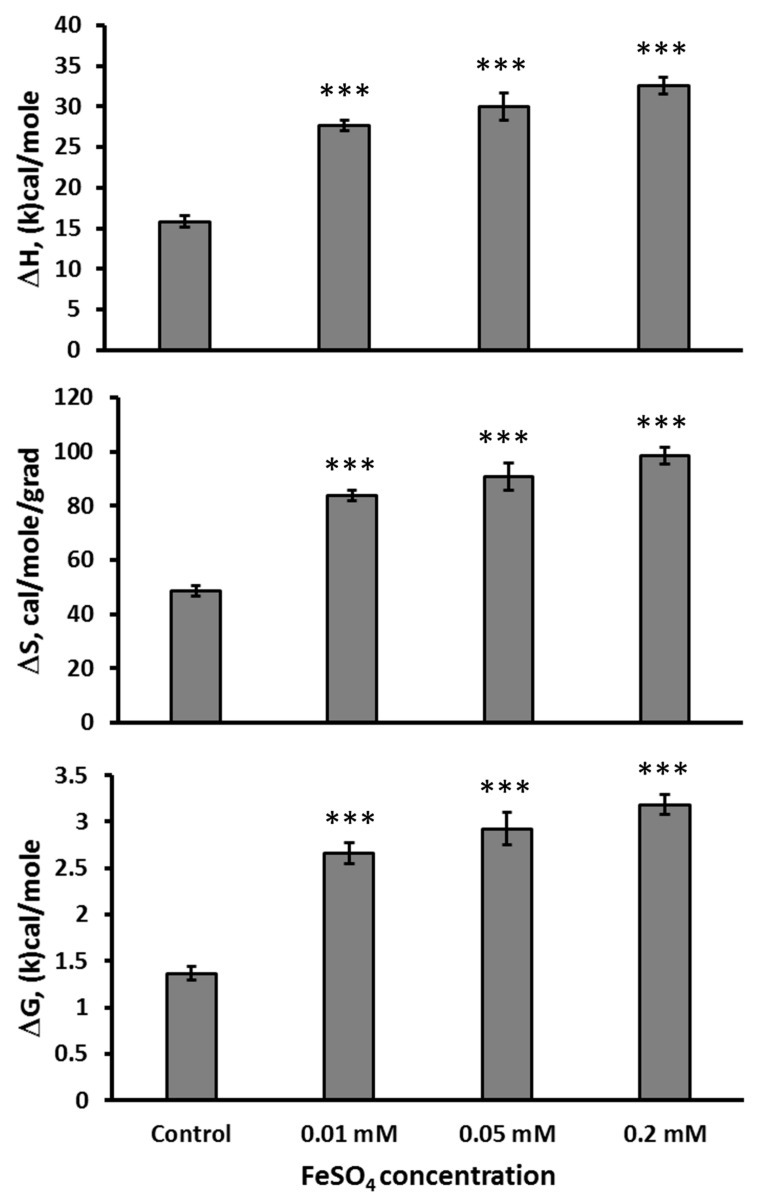
ΔH, ΔS, and ΔG in the thermal denaturation of the mutant 447R form of TPH2 extracted from the midbrain of Balb/c in the absence (control) and the presence 0.01, 0.05, and 0.2 mM of FeSO_4_ in vitro. *** *p* < 0.001 vs. control.

**Figure 2 ijms-26-08188-f002:**
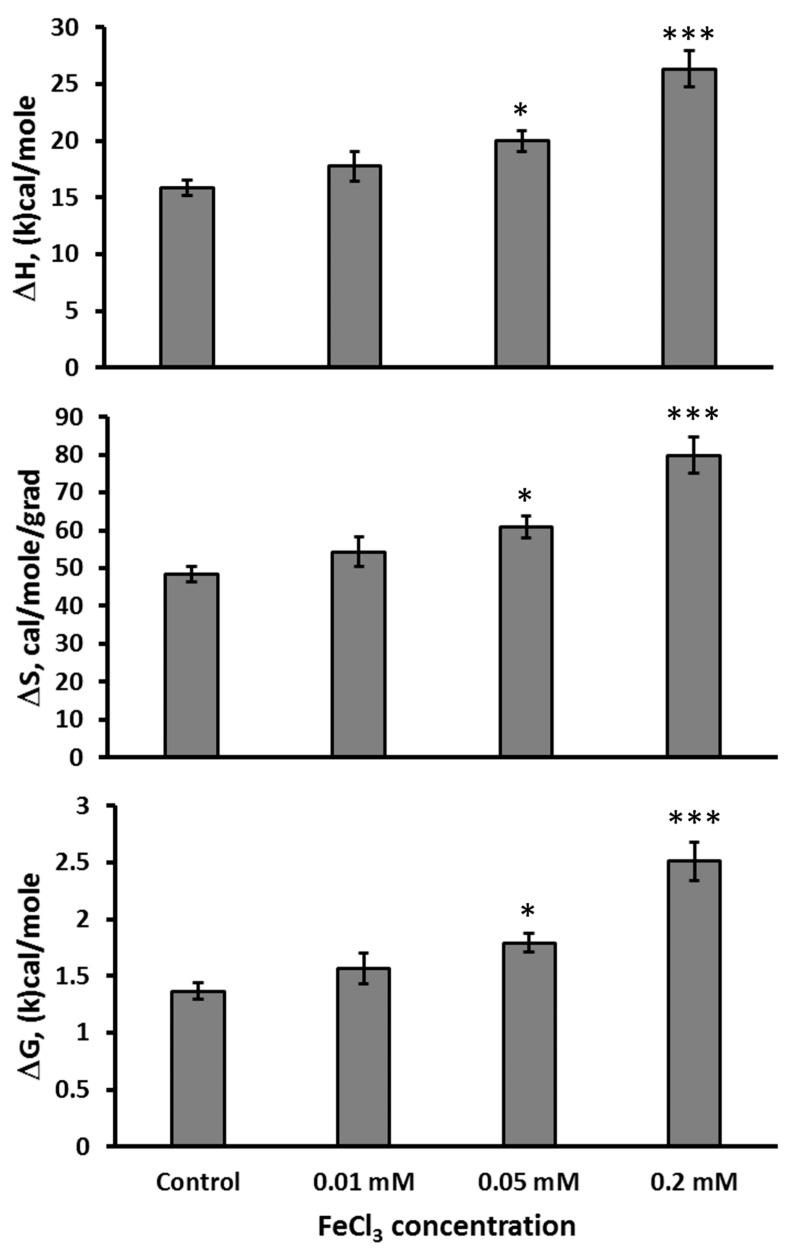
ΔH, ΔS, and ΔG in thermal denaturation of the mutant 447R form of TPH2 extracted from the midbrain of Balb/c in the absence (control) and the presence 0.01, 0.05, and 0.2 mM of FeCl_3_ in vitro. * *p* < 0.05, *** *p* < 0.001 vs. control.

**Figure 3 ijms-26-08188-f003:**
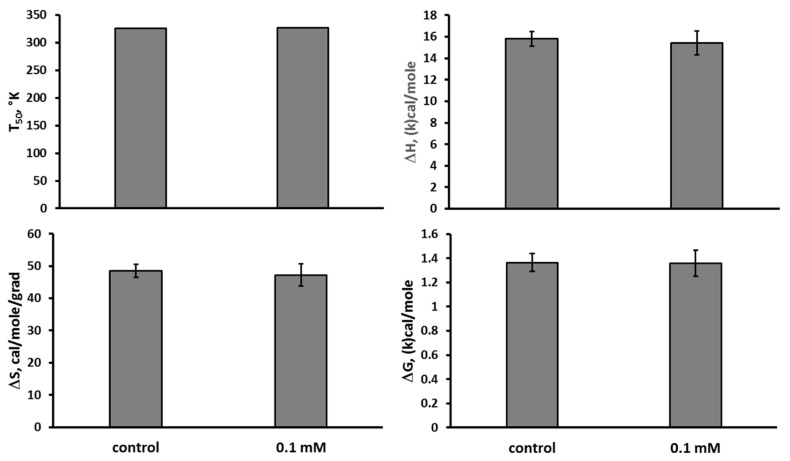
T_50_, ΔH, ΔS, and ΔG in the thermal denaturation of the mutant 447R form of TPH2 extracted from the midbrain of Balb/c in the absence (control) and the presence 0.1 mM of Fe(III) hydroxide dextran complex in vitro.

**Figure 4 ijms-26-08188-f004:**
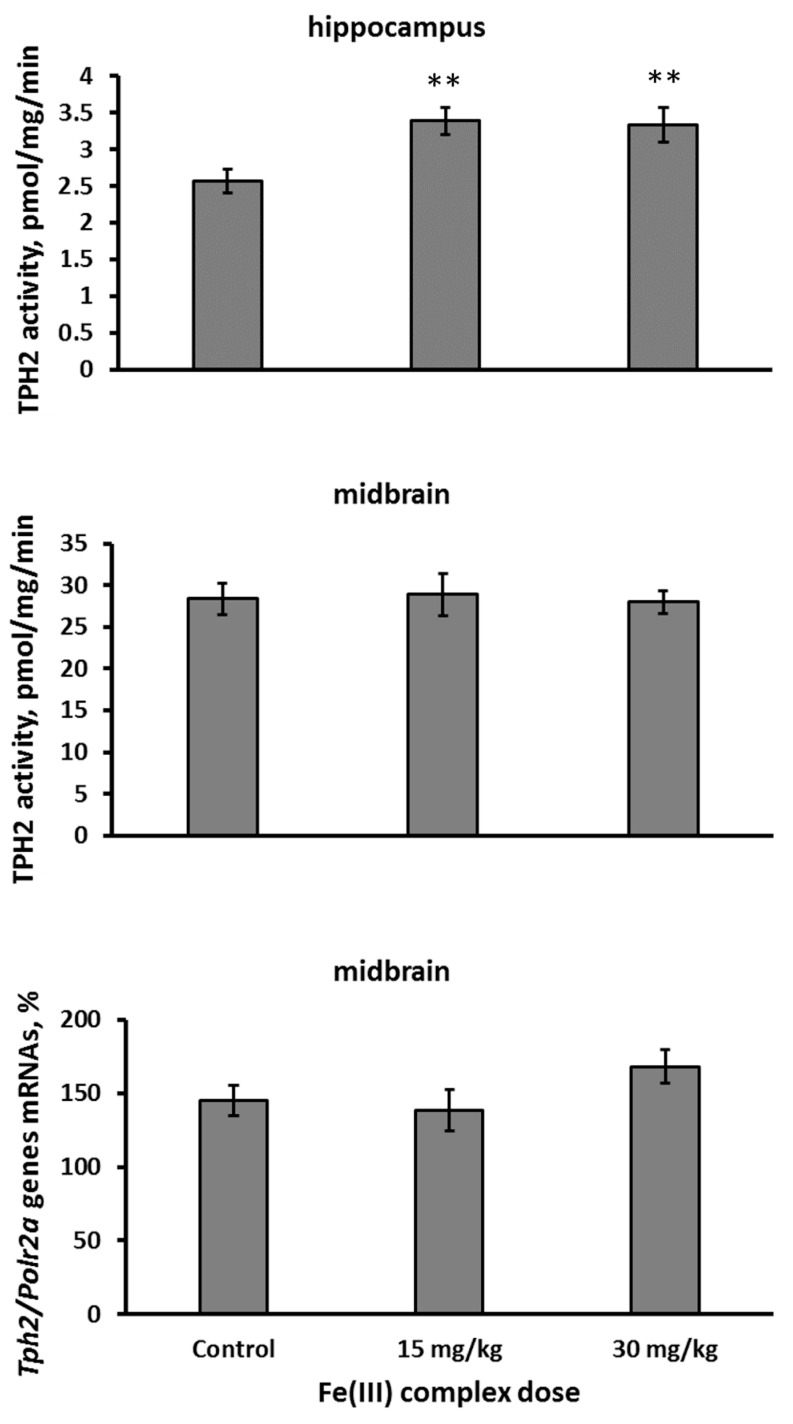
Activity of TPH2 in the hippocampus and midbrain as well as the level of *Tph2* gene mRNA in the midbrain of Balb/c mice after 7 days of intramuscular administration with saline (control), 15 and 30 mg/kg of Fe(III) hydroxide dextran complex.** *p* < 0.001 vs. control.

**Figure 5 ijms-26-08188-f005:**
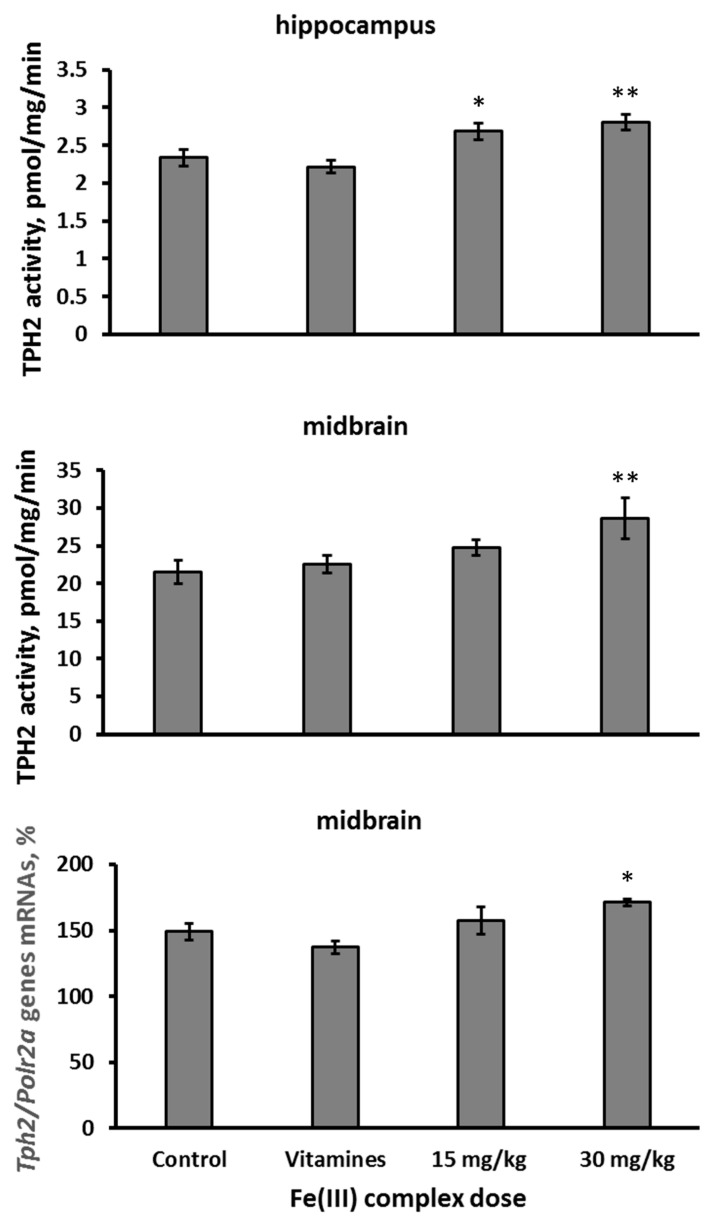
Activity of TPH2 in the hippocampus and midbrain as well as the level of *Tph2* gene mRNA in the midbrain of Balb/c mice after 7 days of intramuscular administration with saline (control), mix of 8 mg/kg of thiamine, 0.08 mg/kg of cyanocobalamin (vitamins), or 15 and 30 mg/kg of Fe(III) hydroxide dextran complex together with this mix of vitamins. * *p* < 0.05, ** *p* < 0.001 vs. control.

**Table 1 ijms-26-08188-t001:** Two-way ANOVA of variability in ΔΔH, ΔΔS, and ΔΔG in the presence of various concentration of two iron ions in vitro.

Parameter	Iron Ion Type	Concentration	Interaction
ΔΔH	F(1,27) = 61.7, *p* < 0.001	F(2,27) = 13.1, *p* < 0.001	F(2,27) = 1.31, *p* < 0.29
ΔΔS	F(1,27) = 61.3, *p* < 0.001	F(2,27) = 13.0, *p* < 0.001	F(2,27) = 1.28, *p* < 0.30
ΔΔG	F(1,27) = 62.3, *p* < 0.001	F(2,27) = 12.8, *p* < 0.001	F(2,27) = 1.50, *p* < 0.24

ΔΔH, ΔΔS, and ΔΔG mean the differences between these parameters in the presence of iron ions and the mean of their control values (without iron ions). The type of iron ions are Fe(II) or Fe(III).

**Table 2 ijms-26-08188-t002:** Sequences, annealing temperatures of the primers, and the size of PCR products (amplicons).

Gene	Sequence	Annealing Temperatures, °C	Amplicon Size, bp
*Polr2a*	5′-TGACAACTCCATACAATGC-3′ 5′-CTCTCTTACTGAATTTGCGTACT-3′	60	194
*Tph2*	5′-CATTCCTCGCACAATTCCAGTCG-3′ 5′-AGTCTACATCCATCCCAACTGCTG-3′	62	239

## Data Availability

The data presented in this study are available on request from the corresponding author. The data are not publicly available due to privacy or ethical restrictions.

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
