# Peer review of "Iron Ions Increase the Thermal Stability In Vitro and Activity In Vivo of the 447R Mutant Form of Mouse Tryptophan Hydroxylase 2"

_ijms, 2025, doi:10.3390/ijms26178188_

Round 1

Reviewer 1 Report

Comments and Suggestions for Authors

This article elaborates on the effects of iron ions on thermal stability in vitro and the activity in the brain of the mutant Tryptophan hydroxylase 2. Since some practical issues need to be addressed before publication, I suggest minor revisions.

  1. The subtitles comment 1 and 2 can be modified with suitable terms.
  2. The resolutions of the images should be improved following MDPI guidelines. Especially, the X and Y axis labels and other labels are not clear.
  3. Explain the significance of measuring Tryptophan 3 Hydroxylase 2 in different brain tissues.
  4. The conclusions can be represented in a single paragraph.

Author Response

This article elaborates on the effects of iron ions on thermal stability in vitro and the activity in the brain of the mutant Tryptophan hydroxylase 2. Since some practical issues need to be addressed before publication, I suggest minor revisions.

The subtitles comment 1 and 2 can be modified with suitable terms.

Answer: We removed the words “comment 1” and “comment 2” (lines 268-271).

The resolutions of the images should be improved following MDPI guidelines. Especially, the X and Y axis labels and other labels are not clear.

Answer: All figures were redrawn and their resolution was increased up to 600 pixels/inch.

Explain the significance of measuring Tryptophan 3 Hydroxylase 2 in different brain tissues.

Answer: We explained our choice of the hippocampus and midbrain (lines 63-64, 266-267).

The conclusions can be represented in a single paragraph.

Answer: We reduced the numbers of items in the Conclusion down to two items.

We have marked all our corrections in red.

We thank you very much for your valuable comments.

Reviewer 2 Report

Comments and Suggestions for Authors

The manuscript “Iron Ions Increase the Thermal Stability In Vitro and Activity In Vivo of the 447r Mutant Form of Mouse Tryptophan 3 Hydroxylase 2” presents a study concerning the both effect of Fe(II) and Fe(III) ions on the energy (DH, DG) of thermal denaturation in vitro, and of repeated administration of Fe(III) hydroxide dextran complex on the activity of mutant TPH2 in mouse brain (in vivo).  My overall comment is that these data present interest concerning the importance of finding a proper pharmacological chaperones for the treatment of inherited diseases caused by mutation misfolding of some proteins. I therefore recommend publishing after minor clarifications and corrections such as:

  • It is known that Fe(II) ions have reducing ability and thus generate ROS. I wonder if it was preserved the oxidation state of Fe(II) during the study or this was changed at Fe(III)? How was monitored this aspect? Because, in mine opinion “the absence of statistical effect of the interaction of iron ion type x concentration on the thermodynamic characteristics of TPH2 thermal denaturation” could result from the Fe(II) into Fe(III) oxidation during the experiment.
  • Perhaps the determination of the iron level in the midbrain and hippocampus of mice toghether with TPH2 activity and the Tph2 gene expression could explain some of hypothesis of the authors.
  • The “can form dative bonds with” must be replaced by “can coordinate to” and from expression “glutamate acid and aspartate acid”, acid must be removed.
  • The T50 notation must be explained.
  • There are some typos that must be corrected (i.e. ctivity (row 16), in vitro and in vivo in Italic style, “in vitro and” expression must be finished (row 39), * must be replaced by · in the compounds formulas).
  • In the sentences that starts with FeCl3 and Fe(III) hydroxide dextran, The must be added at the beginning.
Comments on the Quality of English Language

There are some typos that must be corrected but the Quality of English Language is acceptable.

Author Response

The manuscript “Iron Ions Increase the Thermal Stability In Vitro and Activity In Vivo of the 447r Mutant Form of Mouse Tryptophan 3 Hydroxylase 2” presents a study concerning the both effect of Fe(II) and Fe(III) ions on the energy (DH, DG) of thermal denaturation in vitro, and of repeated administration of Fe(III) hydroxide dextran complex on the activity of mutant TPH2 in mouse brain (in vivo).  My overall comment is that these data present interest concerning the importance of finding a proper pharmacological chaperones for the treatment of inherited diseases caused by mutation misfolding of some proteins. I therefore recommend publishing after minor clarifications and corrections such as:

It is known that Fe(II) ions have reducing ability and thus generate ROS. I wonder if it was preserved the oxidation state of Fe(II) during the study or this was changed at Fe(III)? How was monitored this aspect? Because, in mine opinion “the absence of statistical effect of the interaction of iron ion type x concentration on the thermodynamic characteristics of TPH2 thermal denaturation” could result from the Fe(II) into Fe(III) oxidation during the experiment.

Perhaps the determination of the iron level in the midbrain and hippocampus of mice toghether with TPH2 activity and the Tph2 gene expression could explain some of hypothesis of the authors.

Answer: We agree with this our comments. This problem is more complex and maybe there is an equilibrium between Fe(II) and Fe(III) ions in the incubation environment. In nearest future we plan to reexamine these in vitro results using recombinant TPH2 proteins (see lines 152-155).

The “can form dative bonds with” must be replaced by “can coordinate to” and from expression “glutamate acid and aspartate acid”, acid must be removed.

Answer: These errors were corrected (lines 157-158).

The T50 notation must be explained.

Answer: We explained this notation (lines 48-50).

There are some typos that must be corrected (i.e. ctivity (row 16), in vitro and in vivo in Italic style, “in vitro and” expression must be finished (row 39), * must be replaced by · in the compounds formulas).

Answer: These and some other typos were corrected.

In the sentences that starts with FeCl3 and Fe(III) hydroxide dextran, The must be added at the beginning.

Answer: We added “the”.

Comments on the Quality of English Language

There are some typos that must be corrected but the Quality of English Language is acceptable.

We have marked all our corrections in red.

We thank you very much for your valuable comments.